# EconBERTa: Towards Robust Extraction of Named Entities in Economics

**Karim Lasri**[α,β] **Pedro Vitor Quinta Da Castro**[α,γ] **Mona Schirmer**[α,δ]
**Luis Eduardo San Martin**[α] **Linxi Wang**[α,ε] **Thomáš Dulka**[α] **Haaya Naushan**[α]
**John Pougue Biyong**[α,ζ] **Arianna Legovini**[α] **Samuel P. Fraiberger**[α,ε,η]

[α]The World Bank   [β]Ecole Normale Superieure   [γ]Universidade Federal de Goiás
[δ]University of Amsterdam   [ε]Massachusetts Institute of Technology
[ζ]University of Oxford   [η]New York University

{klasri, pquintadecastro, mschirmer, lsanmartin, lwang34, tdulka,
hnaushan, jpouguebiyong, alegovini, sfraiberger}@worldbank.org

## Abstract

Adapting general-purpose language models has proven to be effective in tackling downstream tasks within specific domains. In this paper, we address the task of extracting entities from the economics literature on impact evaluation. To this end, we release `EconBERTa`, a large language model pretrained on scientific publications in economics, and `ECON-IE`, a new expert-annotated dataset of economics abstracts for Named Entity Recognition (NER). We find that `EconBERTa` reaches state-of-the-art performance on our downstream NER task. Additionally, we extensively analyze the model's generalization capacities, finding that most errors correspond to detecting only a subspan of an entity or failure to extrapolate to longer sequences. This limitation is primarily due to an inability to detect part-of-speech sequences unseen during training, and this effect diminishes when the number of unique instances in the training set increases. Examining the generalization abilities of domain-specific language models paves the way towards improving the robustness of NER models for causal knowledge extraction.

## 1 Introduction

Implementing robust systems capable of automatically extracting structured information from unstructured text is critical for a variety of applications, ranging from event detection (Nguyen and Grishman, 2015; Bekoulis et al., 2019; Tong et al., 2020; Pouran Ben Veyseh et al., 2021; Liang et al., 2022) to building knowledge databases (Khoo et al., 2000; Kim et al., 2020; Harnoune et al., 2021; Wang et al., 2022). In particular, Named Entity Recognition (NER) is a canonical information extraction task which consists of detecting text spans and classifying them into a predetermined set of entity types (Tjong Kim Sang and De Meulder, 2003; Lample et al., 2016a; Chiu and Nichols, 2016; Ni et al., 2017). In the past decade, numerous benchmark datasets have enabled researchers to compare and improve the performances of NER models within specific domains such as science (Luan et al., 2018), medicine (Jin and Szolovits, 2018), law (Au et al., 2022), finance (Salinas Alvarado et al., 2015), and social media (Ushio et al., 2022); in some cases, these datasets have spanned multiple domains (Liu et al., 2020b) or languages (Tjong Kim Sang and De Meulder, 2003). Such datasets are crucial for building models capable of handling a wide range of downstream applications. In medicine for instance, information extraction systems can allow clinician and medical researchers to easily determine which medication is an effective treatment for a specific disease. Analogously, knowing which policy intervention produces a certain economic outcome is imperative for evidence-based policymaking. However, while much work has been done on identifying relevant causal entities in biomedical research (Chang et al., 2022), this question has been unexplored for impact evaluation (IE) in economics. To fill this gap, we introduce `ECON-IE`, a NER dataset of 1,000 abstracts from economic research papers annotated for entities describing the causal effects of policy interventions: *intervention*, *outcome*, *population*, *effect size*, and *coreference*. To the best of our knowledge, this is the first dataset of this kind in economics, thereby laying foundations for causal knowledge extraction in the field.

The introduction of pretrained language models, sometimes referred to as *foundation models* (Bommasani et al., 2022), had a profound impact on the field of information extraction, enabling substantial gains in performance across domains by fine-tuning these general-purpose systems on downstream tasks (Alt et al., 2019; Papanikolaou et al., 2022). In this paradigm, such models are often fine-tuned and deployed to perform a domain-specific task, introducing a distribution shift between the pretraining data and examples seen at inference time. Existing work shows that pretrain-

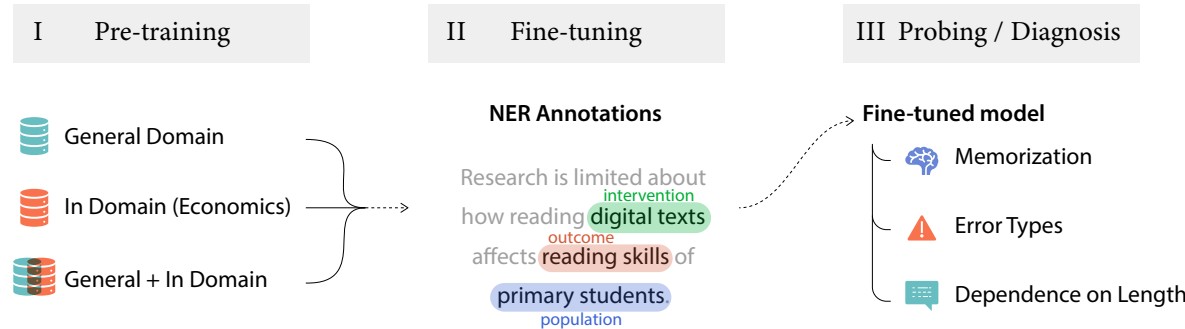

Figure 1: Illustration of the pipeline for the models under investigation, from modeling to diagnosis

ing a language model on documents from the same domain as the downstream task improves performance (Beltagy et al., 2019), where this domain adaptation can either be implemented by in-domain pretraining from scratch (Chu and Wang, 2018; Rietzler et al., 2020; Grangier and Iter, 2022; Huang et al., 2019), or by further in-domain pretraining starting from the weights of an existing pretrained general-purpose model (Lee et al., 2019; Alsentzer et al., 2019). While pretrained language models are available across a wide range of domains (Beltagy et al., 2019; Lee et al., 2019; Chalkidis et al., 2020), economics has yet to follow suit. Therefore we contribute EconBERTa, a language model pretrained on 1.5 million scientific papers from economics, demonstrating that it outperforms open-source general-purpose pretrained models on the ECON-IE benchmark dataset.

While assessing the quality of a model is often achieved by measuring performance after fine-tuning, the knowledge captured by such transfer learning approaches is opaque. Therefore, it remains unclear, which aspects of a model require improvement in order to increase robustness during deployment. Understanding the weaknesses of state-of-the-art models on downstream tasks requires operationalizing the notion of *generalization*. However, we lack a precise definition of generalization in spite of recent attempts to re-explore the concept (Hupkes et al., 2023). For instance, high performance on a held-out test set does not necessarily imply that a model is achieving robust generalization (McCoy et al., 2019). In this context, we challenge the surface performance of our pretrained model, and seek cases where it fails to generalize on the NER task. Our examination reveals three common error patterns. First, we show that most errors occur when the model detects an

entity but fails to properly delimit its boundaries. We further find that, as the number of tokens in a span increases, the model is more likely to incorrectly predict boundaries. Lastly, our model extrapolates more easily to part-of-speech sequences seen during training, a tendency exacerbated by a low number of unique spans for a given entity type in the training dataset.

Taken together, our contributions can be summarized as follows:

- In §3.1, we introduce ECON-IE, a new NER dataset of $1,000$ abstracts from economic research annotated for entities describing the causal effects of policy interventions.

- In §3.2, we present EconBERTa, a language model pretrained on economics research papers, which demonstrates state-of-the-art performance on in-domain NER.

- In §4, we evaluate the generalization capabilities of our fine-tuned NER-model by performing a series of diagnostic tests.

The pretrained EconBERTa models, the ECON-IE dataset, the metadata of the economic pretraining corpus and the source code to replicate the experiments are available via https://github.com/worldbank/econberta-econie

## 2 Related Work

### 2.1 NLP for economics

Existing work in NLP for economics have focused on developing models and datasets for the finance domain. Notably, several studies investigate the pretraining of BERT-based (Devlin et al., 2018) architectures on financial data (Araci, 2019; Yang et al., 2020; Peng et al., 2021; Sakaji et al., 2022).

Previously released datasets typically comprise news articles or press releases annotated for sentiment analysis (Malo et al., 2014), event extraction (Jacobs and Hoste, 2022; Lee et al., 2022; Han et al., 2022), opinion analysis (Hu and Paroubek, 2021) or causality detection in finance (Mariko et al., 2020). In addition to news articles, corporate reports (Loukas et al., 2021; Händschke et al., 2018) are text corpora from economics and business, but lack token-level annotations. Our work fills two gaps by (i) being the first to address information extraction from scientific economic content and releasing a neural language model pretrained in that domain and (ii) defining a NER annotation scheme and releasing an annotated dataset for causal entities of economic impact evaluation.

## 2.2 Domain adaptation

Previous work has also highlighted the utility of domain adaptation when developing models so they can handle tasks in specific domains, as the distribution of input texts can be different in such cases. This has given rise to a number of adaptation techniques (Daumé III, 2007; Wiese et al., 2017; Ma et al., 2019; Cooper Stickland et al., 2021; Grangier and Iter, 2022; Ludwig et al., 2022). In the pretrain-fine-tune paradigm, for pretrained models to generalize over a task in a specific domain, it is advised to fine-tune them on domain-specific datasets, which requires domain-specific annotated resources (Tsatsaronis et al., 2015; Zhu et al., 2022; Au et al., 2022; Li et al., 2021). In this paper, we test whether in-domain pretraining improves performance on a domain-specific task, but we additionally try to gain a better understanding on these models' weaknesses by examining their generalization abilities.

## 2.3 Diagnosing fine-tuned models

Since the rise of general purpose pretrained model, examining the generalization abilities of fine-tuned models has sparked interest in the NLP community. Early work has shown that it is possible to achieve seemingly high performance without learning a given task, and by relying on heuristics and spurious correlations (McCoy et al., 2019; Niven and Kao, 2019; Zellers et al., 2019; Xu et al., 2022; Serrano et al., 2023). This has pushed recent efforts to complement coarse-grained metrics such as accuracy and F1-score with new evaluation sets and metrics (Ribeiro et al., 2020), or with novel ways to diagnose a model's errors (Wu et al., 2019; Bernier-

Colborne and Langlais, 2020; de Araujo and Roth, 2023). In this study, we also propose novel fine-grained analyses aimed at gaining a deeper understanding of our model's performing abilities.

## 3 Experimental Setup

In this section, we introduce the ECON-IE dataset for NER (§3.1), present the training procedure for EconBERTa, and describe the baseline models (§3.2).

### 3.1 ECON-IE dataset

**Data collection**   ECON-IE consists of $1,000$ abstracts from economics research papers, totalling more than $7,000$ sentences.[1] The abstracts summarize impact evaluation (IE) studies, aiming to measure the causal effects of interventions on outcomes by using suitable statistical methods for causal inference. The dataset is sampled from $10,000$ studies curated by 3ie[2] (White and Gaarder, 2009), published between 1990 and 2022, and covering all 11 sectors defined by the World Bank Sector Taxonomy (Bumgarner, 2017). To maximize the diversity of our annotated sample, we perform stratified sampling, i.e. we assign higher sampling weights to sectors and year ranges under-represented in the 3ie database.[3] We then generate a fixed heldout test set by sampling 20% of the abstracts; the remaining 80% are split to perform a 5-fold cross-validation set.

**Annotation procedure**   As our goal is to extract information relevant to describe causal analyses performed in impact evaluation studies, we draw from annotation schemes for clinical randomized control trials. In evidence-based medicine, the PICO elements are a common knowledge representation structure and consists of *population*, *intervention*, *comparators* and *outcome* (Huang et al., 2006). We further adjust the PICO scheme to represent the causal knowledge found in impact evaluation studies. We first follow Nye et al. (2018) and collapse comparators and intervention into a single category. We further extract two additional entity types, namely *effect size* as this information is key to compare the effectiveness of various interventions on an outcome, and *coreference* as spans referring to a previously mentioned entity can be useful for a downstream relation extraction model.

---

[1]See Fig. 1 for an example of a sentence.
[2]International Initiative for Impact Evaluation
[3]See Fig. 6 in App. A.2.6

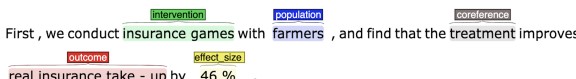

First , we conduct insurance games with farmers , and find that the treatment improves

real insurance take - up by 46 %  .

Figure 2: Example annotation in INCEpTION interface

Based on these entity types, we designed explicit annotation guidelines paired with examples.[4] 10 annotators with at least graduate-level education in economics and experience reading research papers were recruited to perform the annotation task. They received training on the annotation guidelines including videos on how to use the annotation platform INCEpTION (Klie et al., 2018). Each annotator performed a test round on 10 abstracts and received personal feedback from the domain experts who defined the annotation guidelines. We required two rounds of annotation per abstract to ensure an acceptable level of consistency for a reasonable budget. The corpus of abstracts was divided into 20 subsets of 50 abstracts; each annotator was assigned four subsets of 50 abstracts, such that all 1,000 abstracts were annotated by two annotators. Fig. 2 shows an example sentence of resulting annotations.

Given the complexity of the documents and of the annotation task, we expected some level of disagreement between human annotators. We indeed found the inter-annotator agreement measured by the F1-score and the Cohen's Kappa to be equal to $F1 = 0.87$ and $\kappa = 0.71$ respectively. Starting from the annotated abstracts, the experts who defined the guidelines then performed a curation phase to resolve any disagreement between annotators. When a disagreement was not due to a mistake in applying the rules but rather to multiple plausible annotations given existing rules[5], the case was raised and discussed between the experts who came up with a new rule, e.g. *"Annotate the shortest possible span as long as no information is lost"*. The curation phase led to a final annotation per abstract and Table 1 summarizes our dataset statistics after the task was completed.

## 3.2 Models

**Domain-specific language models for economics** While the past few years have seen an explosion of the size of generative language models (Brown et al., 2020; Zhang et al., 2022; OpenAI, 2023), their autoregressive nature makes them less

| # abstracts | 1,000 |
| # sentences | 7,522 |
| # intervention | 6,157 |
| # outcome | 8,628 |
| # effect size | 1,445 |
| # population | 4,824 |
| # coreference | 2,518 |
| Total annotations | 23,572 |

Table 1: Summary statistics of the `Econ-IE` dataset

suited than their bidirectional counterparts for sequence labeling tasks such as NER (Devlin et al., 2018; Schweter and Akbik, 2020; Yamada et al., 2020). The `DeBERTa-v3-base` architecture is one of the current state-of-the-art bidirectional architectures for NER tasks among encoder-based models of the same size (He et al., 2023). Our initial choice was to further pretrain the english-specific `DeBERTa-V3-base` model, however its generator model is not publicly available. We therefore pretrain our economics-specific models based on the `mDeBERTa-v3` architecture by following the ELECTRA pretraining approach (Clark et al., 2020)[6]. Our first model, `EconBERTa-FS`, is pretrained from scratch using a vocabulary tailored to the economics domain. The second one, `EconBERTa-FC`, relies on `mDeBERTa-V3-base`'s generator and discriminator checkpoints as initial weights.

**Pretraining dataset** We pretrain the two `EconBERTa` models on a corpus consisting of 1.5 million economics papers – $800,000$ full articles and $700,000$ abstracts. The documents were collected from the leading digital libraries in the field. A breakdown of the data sources can be found in Table 2. The resulting dataset consists of $9.4$ billion tokens, i.e. almost three times more than the $3.3$ billion tokens used to train BERT (Devlin et al., 2018). For more details about the pretraining procedure, see App. B.1 and App. B.2.

**Baseline models** To evaluate the effects of pretraining the `EconBERTa` models in domain, we compare them against `mDeBERTa-v3-base` (He et al., 2023), which has the same architecture. Additionally, we also include `BERT-base-uncased` (Devlin et al., 2018) and `RoBERTa-base` (Liu et al., 2019) as baselines, as they were extensively studied to analyse the role of domain-specific pretraining (Beltagy et al., 2019; Chalkidis et al., 2020; Nguyen et al., 2020; Carrino et al., 2022).

---

[4]For more details, see App. A.2.6

[5]Examples of sentences where multiple annotations are plausible are presented in App. A.3.

[6]https://github.com/microsoft/DeBERTa

| Source | Documents | Tokens | Type |
|--------|-----------|--------|------|
| RePEc | 399,708 | 5.0B (53%) | full articles |
| SSRN | 355,881 | 3.9B (42%) | full articles |
| NBER | 28,456 | 0.3B (3%) | full articles |
| EconLit | 16,517 | 0.1B (1%) | full articles |
| Scopus | 700,340 | 0.1B (1%) | abstracts |
| Total | 1,500,902 | 9.4B | - |

Table 2: Summary statistics for each source of economics research papers included in the pretraining corpus

**NER fine-tuning**   Finally, all our fine-tuned NER models rely on a conditional random field (CRF) layer for classification (Lafferty et al., 2001), as it has been shown to improve results for sequential classification tasks (Lample et al., 2016b; Souza et al., 2020). All the trainings were conducted using the *crf_tagger* model from the AllenNLP library[7] (Gardner et al., 2017). The hyperparameters used for fine-tuning are presented in table App. B.3.

# 4   Model Diagnosis and Results

In this section, we present the performance of the different pretrained models on the NER task, and analyze their generalization capabilities.

## 4.1   Downstream performance

To characterize the models' ability to generalize on the NER task, we start by measuring their aggregate performance on the ECON-IE dataset. Each model's F1-score is reported in Table 3. In line with previous studies, we find that EconBERTa-FS, which is pretrained from scratch on economics research papers, outperforms the general-purpose mDeBERTa-v3-base model. This result confirms that in-domain pretraining offers a key advantage for maximizing a model's robustness when deployed on a downstream task. Additionally, in-domain pretraining of EconBERTa-FC from an existing pretrained model also improves performance on the task and produces a small but insignificant gain relative to pretraining from scratch. Finally, we find that mDeBERTa-v3-base outperforms RoBERTa-base, which itself outperforms BERT-base, which confirms the findings from previous studies (Liu et al., 2020a; He et al., 2023) and supports our choice of pretraining from a DeBERTa-based architecture.

---
[7]https://github.com/allenai/allennlp-models/blob/main/allennlp_models/tagging/models/crf_tagger.py

| Model Name | F1-score |
|------------|----------|
| EconBERTa-FC | **0.687** (±0.003) |
| EconBERTa-FS | 0.684 (±0.005) |
| mDeBERTa-V3 (He et al., 2023) | 0.670 (±0.004) |
| RoBERTa (Liu et al., 2020a) | 0.659 (±0.004) |
| BERT (Devlin et al., 2018) | 0.649 (±0.003) |

Table 3: Comparison of models' performance for the NER task on the ECON-IE dataset.

## 4.2   Entity-level evaluation

Previous work has argued that comparing F1-scores on a held-out test set fails to highlight the strengths and weaknesses of the models being tested (Fu et al., 2020). In the context of NER, the F1-score is typically computed at the token-level, which makes it difficult to know if a model is capturing entity boundaries accurately. Instead of considering each token separately, a solution is to measure whether the entire span of an entity is accurately predicted. However, only considering an entity to be correctly predicted when the model detects both the correct label and its boundaries would be too restrictive, as it would fail to distinguish between cases when a model completely misses an entity, when it is confuses an entity type for another, or when it fails to capture the correct boundaries.

To address these issues, we draw inspiration from alternative metrics to evaluate information extraction systems and adapt them to our NER task (Chinchor and Sundheim, 1993). We introduce six fine-grained prediction metrics which allow us to analyze the models' successes and failures at the entity-level. In addition to exact matches, three of the metrics account for answers that are partially correct, as described in Table 4. Finally, we count as Missed Labels (ML) entities that were labeled but not predicted at all by a model, which could reflect undergeneralization; and False Alarms (FA) cases when an entity is predicted by a model but no entity was labeled, which could indicate overgeneralization.

| Metric | Entity Type | Boundaries |
|--------|-------------|------------|
| Exact Match (EM) | Match | Match |
| Exact Boundary (EB) | Mismatch | Match |
| Partial Match (PM) | Match | Overlap |
| Partial Boundaries (PB) | Mismatch | Overlap |

Table 4: Attributes of the various prediction types based on the comparison of the predicted and labeled entity types and span boundaries.

These fine-grained evaluation metrics are presented in Fig. 3. We find a consistent pattern across models: most predictions are either matched exactly or with the right label but inexact boundaries (*Partial Match*); on the contrary, label confusions (*Exact Boundary* and *Partial Boundary*) are very rare. We also find that the models miss less than 10% of entities (*Missed Label*), which indicates that they could be undergeneralizing, especially when presented with entities that were not seen during training. Finally, we find that they mistakenly predict annotations that weren't labeled about 10% of the time (*False Alarm*), which suggests that they could also be overgeneralizing, in particular for entities that could systematically be mapped into a label regardless of the context. In the next subsections, we will test alternative hypotheses to better characterize these generalization failures.

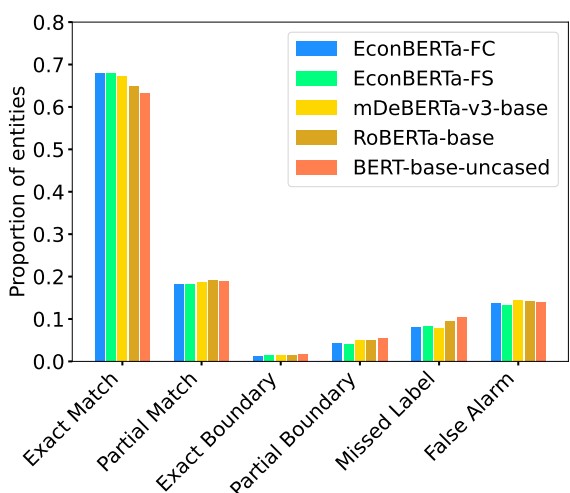

Figure 3: Proportion of exact matches and of the different error types for each model.

## 4.3 Generalization to longer spans

For conciseness purposes, we now focus on EconBERTa-FC for the rest of the analysis, as we demonstrated its ability to outperform other fine-tuned architectures.[8] To test the model's capacity to generalize over *any* entity, we first explore whether the length of an entity affects its ability to detect it accurately. We expect that a longer entity would be harder to predict, as (i) accurately matching boundaries should be more difficult for longer spans, and (ii) longer entities should be less frequent in the

---

[8]The results for rest of the analysis were found to be qualitatively similar across architectures, consistently with Fig. 3.

training data.[9] We therefore examine the impact of lengths on EconBERTa-FC's ability to correctly detect entities in Fig. 4. We find that the model's robustness decreases with entity length. This suggests that the model is failing to generalize properly, as it fails to extrapolate to longer sequences. This failure is relative however, as the reduction in exact matches is mirrored by an increase in partial matches, indicating that the model is still able to capture a sequence which overlaps with the original annotation.

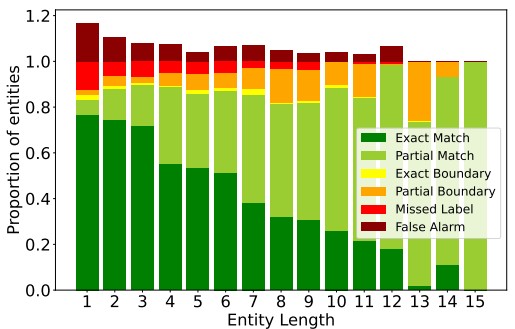

Figure 4: Proportion of exact matches and of the different error types for EconBERTa-FC as a function of entity length.

## 4.4 Inference scheme: reliance on memorization

As our task consists in tagging sub-strings with the correct entity type, a model could rely on a variety of implicit heuristics to achieve good performance without having truly generalized. For instance, we previously identified that our best model was increasingly struggling to correctly capture the boundaries of an entity as span length increases. In this section we test various hypotheses to uncover heuristics causing the model's failures.

**Lexical memorization** We first test the assumption that our model could be memorizing strings that were seen during training to successfully detect entities at test time. In other words, we test whether the model is more successful at detecting *in-train* entities. We compare the predictions at test time in two settings: *in-train$_{lex}$*, in which we isolate entities also present in the training set in the exact same lexical form, and *out-of-train$_{lex}$*, in which we isolate the remaining entities that were not seen

---

[9]Another factor to consider is that the annotation decision on how to set the boundaries of a span becomes increasingly more complex as span length increases.

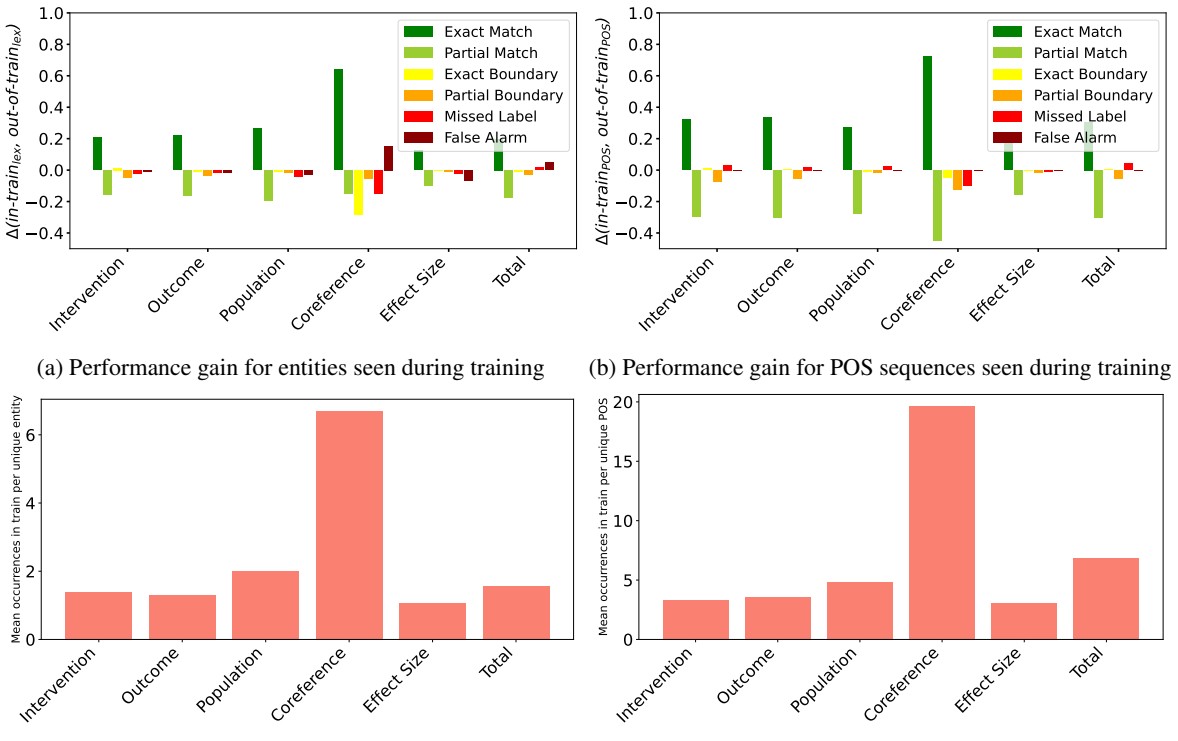

(a) Performance gain for entities seen during training

(b) Performance gain for POS sequences seen during training

(c) Mean count across unique entities in the train set

(d) Mean count across unique POS sequences in the train set

Figure 5: Difference in prediction types for entities and POS sequences that are present and absent from the training set (top) as proxies to measure memorization effects, along with number of unique entities and number of POS sequences (bottom)

during training. We then compute the difference in the proportion of each prediction type between $in\text{-}train_{lex}$ and $out\text{-}of\text{-}train_{lex}$ entities. The results are displayed in Fig. 5a.

We find that, for all entity types, the model produces a higher proportion of exact matches on $in\text{-}train_{lex}$ entities, which suggests a tendency to rely on memorization. This is particularly true for coreferences, where the difference is near 0.6. We also find that the model has fewer partial matches on $in\text{-}train_{lex}$ entities, confirming this observation. For coreferences, we also observe a reduction of mismatched entities, captured by the *"Exact Boundary"* metric, and more false alarms, thus a stronger tendency to overgeneralize by memorizing for this entity class.

We further hypothesize that the mean number of occurrences across unique entities during finetuning could explain the tendency to rely on memorization. As indicated in Fig. 5c, this metric is significantly higher for the *coreference* entity type compared to the other entity types, which suggests a direct link between the tendency for a model to rely on memorization and the mean frequency of occurrence, i.e. the mean count, for each unique

entity present in the training set.

**Lexicosyntactic memorization** In addition to lexical memorization, an alternative hypothesis addressing a milder form of memorization could be that the model learns to associate specific lexicosyntactic sequences seen during training to specific entity types. To test this hypothesis, we first map each entity to its part-of-speech (POS) sequence. We then partition entities at test time between those whose POS is present in the training set ($in\text{-}train_{POS}$) and those whose POS is absent from the training set ($out\text{-}of\text{-}train_{POS}$). For instance, the POS sequence *NNS-NNS* for the *outcome* entity *"household income"* is present in the training set, therefore the *outcome* entity *"school attendance"* which has the same POS belongs to the $in\text{-}train_{POS}$ set. The results of this analysis are presented in Fig. 5b.

We discover that the effect of prior exposure to part-of-speech is higher than prior exposure to lexical content. This suggests that seeing a given POS sequence during training plays a key role in the model's successes and failures. In line with the results on lexical memorization, we find a difference of 0.7 for coreferences on exact matches

(see Fig. 5d). Once again, the differences between entity types in the mean number of occurrences for each unique part-of-speech seems to explain the difference in memorization patterns observed for coreferences. The number of entities for each entity type alone could not explain such effect as Table 1 shows that there are more annotated coreferences than effect sizes in our dataset. Finally, we note that the shares of False Alarms and Missed Labels are not significantly different for $in\text{-}train_{POS}$ and $in\text{-}test_{POS}$ entities, except for coreferences. Again in this case, Missed Labels are fewer for $in\text{-}train_{POS}$, suggesting that the model undergeneralizes and misses entities that are too dissimilar from those seen during training. For other entities, the increase in Exact Match is mirrored by a decrease in Partial Match, which echoes the difference in prediction types for different lengths found in section 4.3.

### 4.5 Disentangling lexical and lexicosyntactic memorization

Finally, we complement the two tests in section 4.4 with a replacement experiment. As we measured a similar effect for both forms of memorization, the root cause for failures remains unknown as both factors are entangled. Behavior tests have been successfully implemented in previous studies to test targeted hypotheses on fine-tuned models (Vajjala and Balasubramaniam, 2022). In our substitution test, we generate random combinations by sampling entities and placing them in contexts where the part-of-speech is preserved. We perform this generation by sampling entities from the training set on the one hand (*in-train*), and from the test set on the other hand (*out-of-train*). If the performance drop is higher in the second case, it shows the model does rely on lexical memorization. If they are similar, the model does not significantly rely on lexical memorization. If the perfomance drop is lower, it is a sign that the model relies on the POS sequence to generalize, but not significantly on the mutual semantic constraints between the context and a given entity, as these factors define the generative procedure for the resulting sentences.

| Condition | △ Acc. | △ Prec. | △ Rec. | △ F1 |
|---|---|---|---|---|
| *in-train* | -0.38% | -1.28% | -2.24% | -1.75% |
| *out-of-train* | -0.39% | -1.28% | -2.56% | -1.92% |

Table 5: Variation in performance when randomly substituting entities based on their lexical category, given that they are sampled randomly from the train set (top) or from the test set (bottom).

We report the results of our substitution tests in Table 5. It disentangles the two forms of memorization (lexical and lexicosyntactic) previously observed. We observe that replacing entities which are identified by a POS sequence present in the training data has little effect over the model's performance overall. This holds true both when the new entity's lexical content is sampled from the training set or out of it. This confirms that the model has reached a degree of generalization that extends beyond the shallow memorization of spans during training. Additionally, the drop in performance is low when performing random replacements, therefore the model's performance does not depend on the mutual semantic constraints between the substituted entity and other parts of the original context to perform the task.

## 5 Discussion

**Memorization: a core component of learning or a sign of undergeneralization?** Throughout this study, we tested several memorization effects ranging from the shallow memorization of lexical patterns to that of lexicosyntactic sequences. While memorization is often opposed to generalization and considered to be a weakness of models (Elango-van et al., 2021; Mireshghallah et al., 2022), its role in the learning process is a source of debate. For instance, some authors have argued that memorization and in particular lexical memorization is a key component of language acquisition and linguistic knowledge (Kess, 1976; Carey, 1978; Nooteboom et al., 2002). While relying too much on memorized associations might be detrimental to learning a task properly (Tänzer et al., 2022) (see results on coreferences in section 4.4), penalizing this phenomenon entirely could also be too prescriptive. Memorization could also play a crucial role either in early or in late stages of the learning process, especially if the task is characterized by a multiplicity of implicit rules and exceptions.

**Learning from mistakes** Uncovering how a model reaches a decision is helpful to understand why it makes certain mistakes and implement modeling strategies to improve its performance. Recent work proposed to teach fine-tuned models to learn from their mistakes on NLU tasks (Malon et al., 2022). Similarly, Xu et al. (2022) attempted to debias NER models by using a counterfactual data-augmentation technique. Together with the diagnoses presented in this paper, these studies pro-

vide a blueprint on how to improve models used for deployment, which constitute interesting directions for future work. To improve models' ability to extrapolate to longer sequences, future work could also explore the impact of the splitting strategy or of augmenting the training data with difficult examples.

## 6  Conclusion

In this work, we presented `EconBERTa`, a novel language model adapted to the economic domain. We demonstrated that it achieves state-of-the-art performance on `Econ-IE`, a novel NER dataset of 1,000 abstracts from economic research annotated for entities describing the causal effects of policy interventions. In doing so, we demonstrated the advantages of domain adaptation, specifically further pretraining from existing checkpoints. Additionally, we diagnosed the model's generalization abilities in a series of fine-grained tests. We found that most mistakes stem from its inability to elicit the exact boundaries of detected entities, and that this phenomenon is due to the model relying on memorization. The memorization effect is mild and extends beyond lexical memorization, as the model is able to extrapolate to entities which have the same part-of-speech sequence as entities seen during training. Finally, we found that the effect of memorization is more pronounced for entities which are less diverse in the training data. Taken together, these findings pave the way towards developing more robust downstream information extraction systems.

## Limitations

The results and methods presented throughout this work present certain limitations which should be taken into account in future work.

**Some of the annotations can be debatable.**  Assigning labels can be straightforward for simple learning tasks, but can become increasingly complex with natural languages due to the multiplicity of strings encountered in real-world corpora and the ambiguity which lies in certain sentences. This could lead to annotations resulting from subjective or even arbitrary judgements from annotators. While we tried to reduce this effect by relying on multiple parallel annotations, arbitrating disagreements, writing precise annotation rules and iterating over the rules to improve their clarity and

coverage, exceptions that are not well captured by our guidelines remain. This problem is recurring for certain annotation tasks documents, limiting the interpretation of performance as the learning task is not suited for the ground truth reference to capture several equally plausible answers.

**Testing generalization abilities requires forming hypotheses.**  In this work, we argued that diagnosing a model's errors is key to improving its robustness on a given task. This evaluation can only be achieved by making assumptions on the mechanisms leading to the errors being observed. Here, we hypothesized that an entity's length and its presence in the training corpus either in its exact lexical form or in a syntactically similar form could affect the ability of the model to detect it. In the general case, forming an hypothesis to diagnose a model's errors is not straightforward, reflecting the fact that there is no consensus on the notion of *generalization*. While future work could draw inspiration from previous studies including ours, there is a long way to go before we obtain a precise characterization of what model generalization truly means and how to operationalize it. The complexity arises from the variety of parameters encountered in a real-world scenario that are affecting a model's ability to extrapolate from its training data.

**Testing a wide range of approaches for in-domain pretraining is computationally expensive.**  During pretraining, each choice of architecture, data selection or preprocessing, and hyperparameters leads to running the entire pipeline, which requires substantial computing resources. Choosing configurations with parsimony reduces the computational load, but it does so at the expense of the range of configurations that can be explored.

## Ethics Statement

The authors foresee no ethical concerns with the work presented in this paper.

## Acknowledgements

We would like to kindly thank Saqib Hussain, Sabrina Yusoff, Leonardo Timperi, Nihaa Sajid, Rohit Tripathi, Nadir Khan, Pierina Forastieri, Morgan Routh for their excellent research assistance.

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

## A ECON-IE Dataset

### A.1 Article Selection

Since our focus lies on causal entity recognition, we limit `ECON-IE` to abstracts from studies dedicated to reveal causal relationships. We keep studies using any of four standard causal inference methods: *randomized control trials*, *difference-in-difference*, *instrumental variable estimation* and *regression discontinuity design*. The 3ie Development Evidence Portal is highly imbalanced across sectors as well as years. By attributing higher weights to under-represented sectors and year ranges, we ensure `ECON-IE` covers different writing styles and sector-specific vocabulary. Fig. 6 displays relative share per sector for the source corpus from 3ie and `ECON-IE`. The latter gives a more balanced coverage of different economic activities.

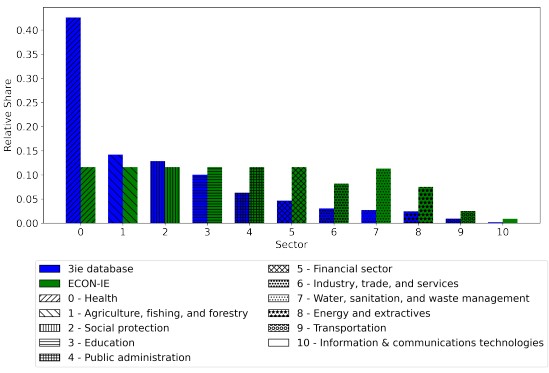

Figure 6: Sector share before (3ie database) and after (`ECON-IE`) article selection process

### A.2 Annotation Guidelines

Our annotation guidelines consist a range of rules, which comprise seven generic rules common to all entity types, and five to eight rules specific to each entity type. In the following, we list the annotation guidelines as presented to the annotators.

### A.2.1 General rules

Span annotation comprises of (1) selecting a span of tokens (determining boundaries of an entity) and (2) tagging it with an entity label. The following rules apply to all entities regardless of their type.

- Use only the context of a sentence for determining entities. The only exception is the coreference, which can refer to interventions or outcomes in other sentences.

- An entity can span over one or more consecutive tokens (words).

- An entity cannot span across sentence boundaries.

- Entity spans cannot overlap.

- Try not to include determiners (the, a), or adjective pronouns (this, its, these, such) in the span.

- Label distributive conjunctions and enumerations as a single span, otherwise as separated spans (e.g. "vitamins A, B, C and D" as a single span but "children and parents" as two).

- Annotate all mentions of entities, including in sentences providing background, conclusions, summaries, or with claims of external validity.

### A.2.2 Intervention

Intervention describes a deliberate involvement in a process or system intended to influence events and/or consequences. In our context, intervention refers to the activities of a project, program, policy, or instrument in the field of development that aims to bring about change in an outcome thereby improving the conditions of a target population. Examples: *conditional cash transfer, leadership programs, household rainwater harvesting, research funding, access to microcredit.*

- Annotate program names and acronyms of interventions as interventions (and not as coreferences), e.g. *Start-and-Improve Your Business (SIYB) Program, No Child Left Behind, Progresa*

- Include generic terms in the intervention span when they describe the entity, e.g. *program, courses, lectures.* For example, in the following sentence, the generic term *courses* should also be part of the intervention span (*business training courses*):

  > *We used a randomized experiment to measure the impact of business training courses.*

  The same applies to other generic terms next to intervention descriptions such as *program* or *intervention*.

- Do not include delivery details or attributes of the interventions. For example, in the following sentence the intervention span should be *Carrying of firearms was banned* excluding the details on which day the ban took place.

> *Carrying of firearms was banned* on weekends after paydays, on holidays and on election days.

- Annotate as interventions mentions of a treatment group by a program description, acronym, or program name; but do not include the term *group* in the span.

- Do not label the standard treatment that the control group receives as an intervention.

### A.2.3 Outcome

An outcome is the likely or achieved change and effects of an intervention. Examples: *adoption of recommended animal breeding practices, institutional trust, math test scores, air pollution, wage levels, HIV treatment success, knowledge of water contamination*

- Include in the span generic references such as the word *effects* or *outcomes* when they are next to an outcome. For example:

  > *The outcome we consider is consumer behavior. Consumer behavior effects* are measured by a household survey.

- Include words that make the outcome a quantity (e.g. *incidence of HIV* and not *HIV*) whenever possible.

- Very important: mentions of effect sizes should not be annotated as outcomes.

- Do not include population in the outcome, e.g. in *teenage fertility, child mortality* tag *teenage* and *child* as population, *fertility* and *mortality* as outcome.

### A.2.4 Population

The population, sometimes referred to as population target, is the group of people on which the intervention is implemented or in which the outcome is measured. Examples: *women and girls, farmers, civil servants, teachers, women in rural areas*

- Include mentions of subgroups of the population where the intervention was applied or on whom an outcome was measured.

- If the text mentions a control group that had a different population and did not receive an

intervention, do not annotate that. We are only interested in the populations that received an intervention.

- Do not annotate standalone generic references such as the word *population* with the population tag.

- Do not label standalone geographical names such as countries, regions or cities. For example, *low-income households in Nairobi*. Label geographical names only when part of greater entity or when otherwise the population wouldn't have a noun. For example, *malnourished Indian children* and *rural Bangladesh*

- Do not label standalone unit of intervention, e.g. *individuals, households, villages*. Label them when part of a greater entity, e.g. *individuals aged 70 years or more, remittance-receiving households*.

- Do not label treated group as population, e.g. *impregnated chaddar users* (*impregnated chaddar* is the intervention; *user* is a generic term and not labeled).

- Words defining the status of the population should be part of the population span, e.g. in *child with acute diarrhea* and *mothers suffering from anemia* the whole spans are population.

### A.2.5 Effect size

Effect sizes are the mentions of quantitative measures of the magnitude of the intervention's effect in an outcome. Examples: *28 percentage points, 4.7%, 0.37 births, 102 μ/m3*

- Whenever possible, the effect size span should include not only the numeric measure but also the unit of measure, as long as this is not part of an outcome span. For example,

  > *PM2.5 exposure means were 266 and 102 μ/m3 during the trial period in the control and intervention groups, respectively.*

- Tag all the mentions of effect sizes, including mentions for different treatment groups, subgroups, control groups, or about the difference between treatment and control groups.

- Do not include in the span words indicating the direction of the effect size (example: *increment, decrease, increase*).

  > *The results indicate that primary school completion reduces teenage fertility by 0.37 births and the incidence of teenage childbearing by around 28 percentage points.*

- Statistical measures such as t-stats, p-values, or confidence intervals should not be considered effect sizes.

- Odds ratios and incidence rate ratios should be considered effect sizes.

### A.2.6 Coreference

A coreference accounts for all generic expressions that relate to the same intervention or outcome entity. Examples: *intervention, project, program, outcome, effect, results, it, they*

- Tag only coreferences of interventions and outcomes.

- Mentions of an intervention by its name (e.g. *Progresa, Sesame Street*) should not be tagged as coreferences but as interventions.

- Words *treatment, intervention, experiment* in *treatment group, intervention group, experiment group* should also be annotated as coreferences excluding the word *group* in the span.

### A.3 Multiple Plausible Annotations

Multiple annotations can be plausible for the same sentence. We illustrate this using examples for which (non-exhaustive) competing options displayed in Table 6.

Our examples comprise candidate nested spans (1, 2), conjunctions of multiple spans (3), auxiliary details (4), enumerations (4, 5, 6), potential risk factors rather than outcomes (6), vague entities (7), multiple populations (8), and nested entities (9).

## B EconBERTa Models

### B.1 Pretraining data collection

We sourced economic articles for pretraining from a variety of online libraries. To ensure economic-specific content in the pretraining corpus, we implemented a two-step article selection process. In the first step, we collect articles from both (i) econ-specific libraries (RePEc, Econlit and NBER) and (ii) the economic category of general science libraries (Scopus and SSRN). Despite this first filtering step, the resulting corpus from the initial stage turned out to be diverse due to the inclusion of articles from neighboring fields such as law and political science.

To ensure we isolate content specifically rooted in economics, we implemented a second filtering process based on the journal in which the article was published. An article is included in the pretraining corpus if its publication venue meets any of the following criteria:

- It is listed among the most influential economics journals by the SCImago Journal and Country Rank (González-Pereira et al., 2009), containing 1166 journals.

- The journal has been classified as economic-specific in a manual annotation process of 1500 journals that we carried.

- The journal name contains the string "econ".

- The paper is a working paper from RePEc.

The second filtering step reduced the corpus size from 2.5 million to 1.5 million documents.

### B.2 Pretraining procedure

The two EconBERTa models developed in this work were pretrained using the ELECTRA-style training (Clark et al., 2020). We display pretraining hyperparameters in Table 7. For EconBERTa-FS, our model pretrained from scratch, we used a SentencePiece vocabulary containing 128.000 tokens, trained on the same pretraining corpus. As for EconBERTa-FC, the mDeBERTa based approach, we used the same vocabulary as DeBERTa-V3-base, containing the same amount of tokens. We opted for using a vocabulary size matching DeBERTa's instead of its multilingual counterpart due to its larger dimension, and adjusted the dimensions at the embeddings layer. The resulting weights were used as an initial checkpoint for EconBERTa-FC. Both models were trained on a cluster composed

| Number | First annotation option | Second annotation option |
| --- | --- | --- |
| 1 | We find that participating in a CSA increased families' likelihood to report having saved money *(outcome)*. | We find that participating in a CSA increased families' likelihood to report having saved money *(outcome)*. |
| 2 | We evaluate one attempt to make local institutions more democratic and egalitarian by imposing participation requirements *(intervention)* for marginalized groups (including women) and test for learning-by-doing effects. | We evaluate one attempt to make local institutions more democratic and egalitarian by imposing participation requirements *(intervention)* for marginalized groups (including women) and test for learning-by-doing effects. |
| 3 | Neither program affected progression to secondary school, but children in grade 6 in SFP schools at baseline were significantly more likely to remain in primary school and repeat a grade *(outcome)* than drop out. | Neither program affected progression to secondary school, but children in grade 6 in SFP schools at baseline were significantly more likely to remain in primary school *(outcome)* and repeat a grade *(outcome)* than drop out. |
| 4 | The nutrition education intervention programme (NEIP) *(intervention)* comprised ten topics emphasising healthy eating, hygiene and sanitation. | The nutrition education intervention programme (NEIP) *(intervention)* comprised ten topics emphasising healthy eating *(intervention)*, hygiene *(intervention)* and sanitation *(intervention)*. |
| 5 | This study has brought about positive changes in the knowledge, attitude and practice of mothers and their children towards the disposal of children's faeces *(outcome)* in rural communities. | This study has brought about positive changes in the knowledge *(outcome)*, attitude *(outcome)* and practice *(outcome)* of mothers and their children towards the disposal of children's faeces in rural communities. |
| 6 | Mothers giving birth to low birth weight babies (LBWBs) have low confidence in caring for their babies because they are often still young and may lack the knowledge *(outcome)*, experience *(outcome)*, and ability to care *(outcome)* for the baby. | Mothers giving birth to low birth weight babies (LBWBs) have low confidence in caring for their babies because they are often still young and may lack the knowledge, experience, and ability to care for the baby. |
| 7 | We find large positive impacts on school enrolment *(outcome)*, number of teachers *(outcome)*, and other inputs *(outcome)* for programme schools near the minimum pass rate. | We find large positive impacts on school enrolment *(outcome)*, number of teachers *(outcome)*, and other inputs for programme schools near the minimum pass rate. |
| 8 | We implement a randomized experiment offering Salvadoran migrants *(population)* matching funds for educational remittances, which are channeled directly to a beneficiary student *(population)* in El Salvador chosen by the migrant *(population)*. | We implement a randomized experiment offering Salvadoran migrants *(population)* matching funds for educational remittances, which are channeled directly to a beneficiary student in El Salvador chosen by the migrant *(population)*. |
| 9 | We find that girls are more likely to refuse free eyeglasses, and that parental lack of awareness of vision problems, mothers' education, and economic factors (expenditures per capita and price) significantly affect whether children *(population)* wear eyeglasses *(outcome)* in the absence of the intervention. | We find that girls are more likely to refuse free eyeglasses, and that parental lack of awareness of vision problems, mothers' education, and economic factors (expenditures per capita and price) significantly affect whether children wear eyeglasses *(outcome)* in the absence of the intervention. |

Table 6: Examples where multiple annotations are plausible

of 16 nodes, each one comprising 4 RTX 8000 GPUs, with 48GB of RAM each. We used the pre-processing available in DeBERTa's github repository.[10]

| Hyper-parameter | EconBERTa-FS | EconBERTa-FC |
|---|---|---|
| Number of Layers | 12 | 12 |
| Hidden Size | 768 | 768 |
| FNN Hidden Size | 3072 | 3072 |
| Attention Heads | 12 | 12 |
| Attention Heads Size | 64 | 64 |
| Dropout | 0.1 | 0.1 |
| Warmup Steps | 10% of total | 10% of total |
| Learning Rate | $7 \times 10^{-4}$ | $6 \times 10^{-4}$ |
| Batch Size | 3600 | 3600 |
| Weight Decay | 0.01 | 0.01 |
| Max Steps | 65,388 | 68,650 |
| Epochs | 10 | 10 |
| Learning Rate Decay | Linear | Linear |
| Adam $\epsilon$ | $1 \times 10^{-6}$ | $1 \times 10^{-6}$ |
| Adam $\beta_1$ | 0.9 | 0.9 |
| Adam $\beta_2$ | 0.98 | 0.98 |
| Gradient Clipping | 1.0 | 1.0 |

Table 7: Hyper-parameters for pre-training

### B.3 Fine-tuning Parameters

We display in Table 8 the hyperparameters used to fine-tune each of our models.

| Hyper-parameter | Value |
|---|---|
| Dropout of Task Layer | 0.2 |
| Learning Rate | [5e-5, 6e-5, 7e-5] |
| Batch size | 12 |
| Gradient Acc. Steps | 4 |
| Weight Decay | 0 |
| Maximum Training Epochs | 10 |
| Learning Rate Decay | Slanted Triangular |
| Fraction of steps | 6% |
| Adam $\epsilon$ | 1e-8 |
| Adam $\beta_1$ | 0.9 |
| Adam $\beta_2$ | 0.999 |

Table 8: Hyper-parameters for fine-tuning

### C  Training Dynamics

In order to test whether domain adaptation impacted significantly the difficulty to learn the task for our pretrained models, we measured how the error types and memorization-related metrics evolved with the number of epochs. We display such results in Fig. 7. While we pretrained our models with 10 epochs (see Table 8, we see that most models almost reach their final share of exact matches after only a few epochs. We further show that the share

of exact match among entities and part-of-speech not seen during training also stabilizes early during finetuning overall. This seems especially true for EconBERTa-FC, our best performing model, while it takes more epoch for BERT, the worst performing model. Note however that the observation is qualitative at this stage and the evidence remains weak.

---

[10]https://github.com/microsoft/DeBERTa/blob/master/experiments/language_model/prepare_data.py

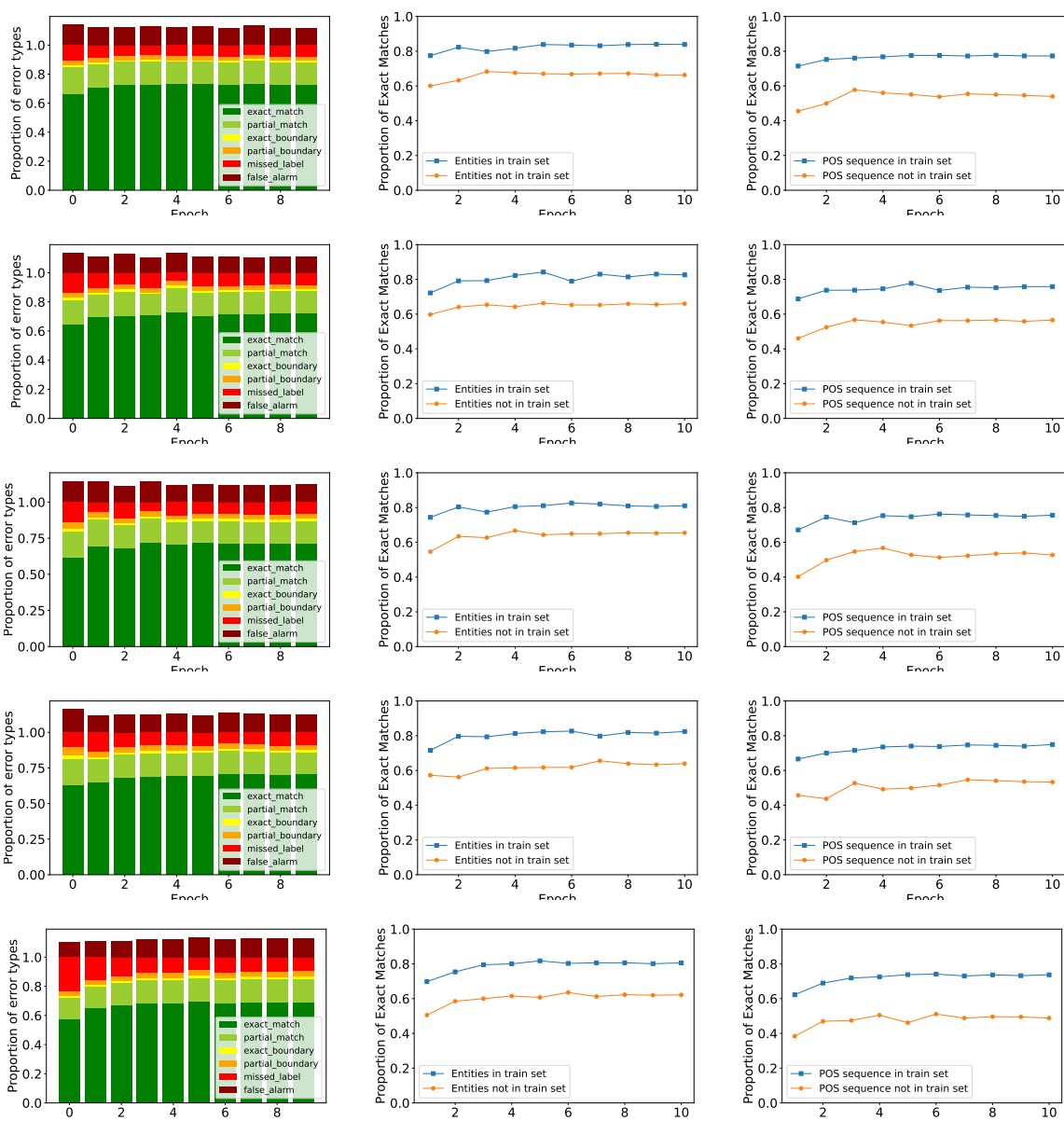

Figure 7: Training dynamics for EconBERTa-FC, EconBERTa-FS, mDeBERTa-v3, RoBERTa and BERT (from top to bottom). In all figures, the x-axis shows epochs. We display the evolution of shares of each prediction type (left column), as well as the rate of Exact Match for entities present/absent from the train set (middle column) and part-of-speech sequences present/absent from the training set (right column).