# OpenReview forum: "EconBERTa: Towards Robust Extraction of Named Entities in Economics"
_EMNLP/2023/Conference — EMNLP 2023 Findings_

### Official Review · Reviewer_Cyad · 2023-07-26

**Soundness:** 4

**Excitement:**

3: Ambivalent: It has merits (e.g., it reports state-of-the-art results, the idea is nice), but there are key weaknesses (e.g., it describes incremental work), and it can significantly benefit from another round of revision. However, I won't object to accepting it if my co-reviewers champion it.

**Paper Topic And Main Contributions:**

The authors of this paper focus on training a LLM (based on DeBERTa) for the economics domain.  They train their new model, EconBERTa, on 1.7 million economic papers.  The authors apply this model to the information extraction task, for which they annotate a new economics-specific dataset.  To test the performance of EconBERTa, they fine-tune a CRF layer for NER classification for EconBERTa and several baseline models.  They find that EconBERTa has a small but significant performance increase over baselines.

The authors then proceed to perform a large evaluation of the model's performance.  This includes finer-grained metrics to evaluate where errors are occurring, including boundary errors or entity type errors.  They find that most errors consist of boundary errors with the correct entity type.  Further, they find that entity length has an effect on performance.  Finally, they probe the model's tendency to memorize, and explore whether it memorizes via lexical memorization or via lexicosyntatic memorization (i.e. part of speech).

The authors promise to release their dataset and code.

**Reasons To Accept:**

1) The paper was well written.  I found it very easy to follow!
2) The authors focus on a less-studied domain (economics), which expands the reach of NLP beyond standard datasets. The authors created a new IE dataset for the domain and a new foundation model.
3) The authors show a performance increase for their target dataset, with statistical testing and reasonable baselines.
4) The paper includes an extensive analysis of where the model performs well and does not perform well.

**Reasons To Reject:**

1) Without diminishing the impact of new datasets within the field, the authors do not introduce any new modeling techniques for the EconBERTa model. They appear to use a standard fine-tuning technique, and the fact that this improves performance on the in-domain NER task is predicted by other work.
2) I'm not sure the content of this paper is sufficient for a long paper.  The model analysis is a great addition to the paper, but I wonder if reconfigured (specifically sections 3.2 - 3.5, the highlights included in the main paper, the details in the appendix), if it isn't better suited for a short paper contribution.  This is especially true since the analysis generally meets expectations of other models.  I think this might make the paper stronger.
3) I think another path to making this paper stronger is delving deeper into the specifics of the domain.  I don't see much discussion of what makes Economics NER distinct in the evaluation section - is this just the same as any other NER dataset?  Even better would be an exploration of modeling changes that are specific to Economics can improve performance, but I understand this is a more difficult task.

**Reproducibility:**

4: Could mostly reproduce the results, but there may be some variation because of sample variance or minor variations in their interpretation of the protocol or method.

**Reviewer Confidence:**

4: Quite sure. I tried to check the important points carefully. It's unlikely, though conceivable, that I missed something that should affect my ratings.

---

> ### Author Rebuttal · Authors · 2023-08-28
>
> We first wish to thank the reviewer for the insightful comments, as well as the comprehensive acknowledgement of our paper’s strengths.
>
> We agree that the novelty in our paper does not lie in the model development pipeline. That being said, we produce a novel dataset which can be a precious resource for information extraction in the fields of economics, and a novel model which outperforms publicly available architectures for NER extraction in that domain. We also introduce novel experiments to diagnose how models generalize after they have been trained, which can inspire future diagnoses of fine-tuned models in a variety of settings for downstream deployment.
>
> Regarding the suggested length of the paper, we agree that our results and contributions would fit in a short paper if we planned on releasing only the dataset and best performing model. However, we believe that our experiments are crucial as diagnosing models is an important quest in NLP, given that surface performance can be deceitful. This has been shown for instance in previous work which we cite, including McCoy, Pavlick and Linzen (2019) along with a large body of research cited in l.462-464.
>
> We also fully agree that it would be very useful to present the specificities of our NER task, as detecting our entities can be less straightforward than detecting other entity types that are more frequent in the literature. For example, disagreement can be rarer for the boundaries of a span describing a person, a location or a quantity. However, for populations or interventions, we fall into some difficulties regarding e.g. the level of details that we want to keep (e.g. “children aged 4-10”). We formulated precise annotation rules to handle such cases, in order to avoid any confusions, for annotators and for the model. However the obtained set of entities are heterogeneous in length and complexity, which makes the task harder for the model than NER addressing simpler entity types. This specificity is not exclusive to economics however, but to the type of entities that we are extracting in this context. We plan on clarifying this in the final version of the paper, and are thankful for this suggestion as it will make the paper stronger, and can be beneficial to a broader audience. Also, please note that the annotation guidelines will be made public along with the ECON-IE dataset.

---

### Official Review · Reviewer_iiaL · 2023-07-29

**Soundness:** 3

**Excitement:**

4: Strong: This paper deepens the understanding of some phenomenon or lowers the barriers to an existing research direction.

**Missing References:**

Related Work: There is a lot of work on domain-adaptive pre-training similar to what is introduced here in the paper. There is more work on domain-adaptive pre-training:
  - https://aclanthology.org/D19-1433 (Han and Eisenstein, 2019)
  - https://aclanthology.org/2020.acl-main.740 (Gururangan et al., 2020)
  - https://aclanthology.org/W19-1909/ (Alsentzer et al., 2019), ClinicalBioBERT
  - https://aclanthology.org/2022.naacl-main.366 (Zhang et al., 2022), JobBERT

The evaluation suite is very similar to SemEval 2013, task 9: https://aclanthology.org/S13-2056.pdf. Possibly reference to this.

**Paper Topic And Main Contributions:**

This paper is about a new language model for the Economy domain: EconBERTa, two models EconBERTa-FC further pre-trained from a checkpoint and EconBERTa-FS pre-trained from scratch on 10.8B tokens with a (m)deBERTa-v3 architecture. They furthermore release a complementary NER dataset ECON-IE to extract named entities from scientific publications of economics. They annotated for 5 classes: intervention, outcome, effect_size, population, coreference. They show that their model EconBERTa-FC works best. They further do analysis on memorization of training, performance on span length, and analyze types of errors.

**Questions For The Authors:**

- Out of curiosity, why is it important that we have a language model for the economy domain? Specifically, why do we annotate for these specific named entities as proposed in the paper and nb economic research papers? I'm missing this connection in the paper.
- Is the pre-training data also consisting of Abstracts from Economics papers? If not, how did you obtain the full text from the PDF e.g., SSRN?
- Is there any chance that you release the full annotations before they were merged? This is also valuable information.
- In the results, the best-performing model has an F1-score of 0.736. In the analysis, we can see a gain of 0.5 (what?) when we measure between in-train and out-train. Is this 50 F1 points or just 0.5 F1 points?

**Reasons To Accept:**

- The paper is well-structured, clearly written, and an interesting domain. In addition, the authors release a novel dataset for the economy domain.

**Reasons To Reject:**

- The annotation process is rather handwavy, there's a lot of information missing:
  - How many rounds did the annotators annotate for?
  - Was there any discrepancy between annotators?
  - What did the annotators get paid?
  - Did they uniformly annotate as much as others?
  - Why two annotators per abstract and not more?
  - In Appendix B3, the authors mention multiple plausible annotations, how did the final annotator decide on which one to choose?
  - What was the overall agreement between annotators?
- The analysis generally could be improved.
  - The sections on memorization are verbose and not motivated (why memorization?). This analysis could be much shorter by calculating the overlap between unique entities of train/test. This already could explain the performance differences between train and test. Given the low number of unique entities in, e.g., coreference, it could already be that the in-train split, could be much larger than the out-of-train split.
  - The authors evaluated with an in-train and out-train split where the entities overlap or do not overlap with test. It is unclear what the performance gains are on the y-axis, is this F1? If so, are these gains of 0.x F1 or xx.0 F1?
  -For the lexicosyntactic memorization, why exactly POS tagging? Why not dependency parsing? Would this have the same results? This is not mentioned.
  - I personally would not suggest doing an analysis on the test set. Insights on the test set can influence model development and thus inflate performance metrics. Therefore, I suggest using a dev. set next time.

**Reproducibility:**

2: Would be hard pressed to reproduce the results. The contribution depends on data that are simply not available outside the author's institution or consortium; not enough details are provided.

**Reviewer Confidence:**

4: Quite sure. I tried to check the important points carefully. It's unlikely, though conceivable, that I missed something that should affect my ratings.

**Typos Grammar Style And Presentation Improvements:**

- There is some inconsistency in usage of a period after a paragraph header;
- In the discussion section, you have a paragraph on "learning from mistakes". Even though you pinpointed the mistakes the model makes in the analysis, you did not leverage this information to improve the model. Personally, a bit of a handwavy paragraph that could be removed.
- I suggest creating a data statement (Bender and Friedman, 2018): https://aclanthology.org/Q18-1041
- The Conclusion section is missing a number.

---

> ### Author Rebuttal · Authors · 2023-08-28
>
> We first wish to thank the reviewer for the insightful comments and questions.
>
> We will clarify the annotation process by incorporating the reviewer’s suggestions:
> * The corpus of abstracts was divided into 20 subsets of 50 abstracts. Each annotator was assigned four subsets of 50 abstracts, such that all 1,000 abstracts were annotated by two annotators. Before producing the final annotations, the annotators first performed a practice round on 10 abstracts each and received feedback from the NLP researchers who designed the guidelines.
> * As mentioned in l. 170-175, there were discrepancies between annotators. We computed IAA and found f1=0.87 and Cohen’s Kappa=0.71 prior to curation.
> * In some cases, the disagreement was due to at least one annotator not properly applying the annotation rules. In other cases where multiple annotations are plausible, we added a new rule after discussion between our NLP and economics experts, e.g. “annotate the shortest span as long as no information is lost”.
> * Our annotation process requires competent annotators (“at least graduate-level education in economics” l. 161-164) which are a costly human resource. We chose to perform two rounds of annotation per abstract to ensure an acceptable level of consistency for a reasonable budget, and included a curator to resolve disagreements.
>
> Additionally, we will strengthen the motivation for our analysis:
> * Memorization (l.331-339) is often opposed to true generalization and the goal with our in-depth analyses is to test if the model truly generalizes (e.g. to long sequences, or spans not seen during training). Assessing generalization in NLP beyond coarse-grained metrics has been a hot topic in NLP over the past years (l. 92-109, l. 456-470, l.486-507).
> * We motivate the importance of an in-domain model in l. 73-91: they allow performance gains on tasks performed in a given domain. Our model is valuable for a wide range of downstream applications in economics including information extraction, and could become as useful as successful counterparts in other domains, such as BioBERT, or BERTTweet. cited in l.85 and l.220-221.
> * We also specify that causal inference analysis is almost ubiquitous in economics literature and the five entities we extract are commonly used to summarize impact evaluation research. This specificity makes the target classes unlikely to be found in the existing NER literature. We already mention that our aim is to extract causal knowledge (l.24, 60, 141, 1183) but will extend on their utility for practitioners in the field of economics and impact evaluation.
> * Additionally, we will clarify that we chose to identify spans using POS sequence as it can be restricted to a subspan for a given entity, while dependency parsing would have arrows pointing to the sentence and is less handy to identify a subsequence syntactically. Another candidate was constituency parsing, which we also discarded as the boundaries of our annotated entities do not necessarily match with those of syntactic constituents.
> * We also shared the intuition that overlap between entities found train/test sets alone could account for memorization effects displayed in fig. 4, but it is not as informative as the mean number of occurrences for unique instances found in the train set: it varies between 61% and 75% for all classes. We can add this in the appendix.
>
> As for the other questions asked:
> * All scores are displayed in a range from 0 to 1 (0.x F1), except from table 5 where the gains displayed are F1 points (xx.0 F1), due to their small magnitude (we report a “little effect” in l.442). We will display them as 0.x F1 in the final version of the manuscript to avoid any confusion and mention it in the text.
> * We fully agree that the test set shouldn’t interfere with model selection. We carry the memorization analysis only after the best performing model was chosen, a process that did not influence model selection. In doing so, we separate this diagnosis analysis from the model development phase. For the final version of the paper, we can display this analysis for the dev set in the appendix.
> * For pretraining, we did use the full text of articles which we obtained from pdfs by performing text extraction using a pdf2txt python library, and further preprocessed the text to remove noise. We then performed a random sampling across sources to control for the quality of our preprocessing. We will add more details on this full preprocessing pipeline in the final draft.
>
> Finally, we thank the reviewer for the missing references and will also add more information about our annotators in the appendix as suggested in Bender and Friedman’s paper, as this can lead to improved transparency and clarity regarding the annotation process.

---

### Official Review · Reviewer_XXaT · 2023-08-10

**Soundness:** 3

**Excitement:**

3: Ambivalent: It has merits (e.g., it reports state-of-the-art results, the idea is nice), but there are key weaknesses (e.g., it describes incremental work), and it can significantly benefit from another round of revision. However, I won't object to accepting it if my co-reviewers champion it.

**Paper Topic And Main Contributions:**

The paper presents a dataset of 1000 economics abstracts for named entity recognition and models trained on economics literature for five types of entities relevant to impact evaluation studies in economics. The paper further analyzes the models' performances based on a set of entity characteristics, such as the entity length and POS tags.

**Questions For The Authors:**

Question 1: Who prepared the annotation guidelines?

Question 2: When the annotations are different, either overlapping or not, how do you resolve them?

Question 3: Did you compute the inter-annotator agreement at any point of annotation process?

Question 4: Do you do any pre-processing of the 1.7 million papers?

Question 5: Isn't the EconBERTa-FS trained from scratch on economics domain? Isn't its performance worse than EconBERTa-FC?
The description in the text is not consistent with the results. lines 242 - 243 and 249 -250

Question 6: How do you achieve the results in Table 3? What data do you train the models on? 80% of 1000 abstracts? what data do you use to validate your model on? What is the distribution of entities in each set?

Question 7: Are the results in Table 3 exact match?

Question 8: What does entity length of "0" mean?

Question 9:  What you exactly mean by isolating entities in section lexical memorization? What happens if in one sentence one entity is in -train and another not?

Question 10: Can you show the performance of the models for each entity?

Question 11: How long did the training of models take?

Question 12: Wasn't EconBiz data (https://www.econbiz.de/) appropriate for the paper's analysis?

**Reasons To Accept:**

- The paper provides a new dataset for NER in economics.
- The paper provides a model for NER in economics.

**Reasons To Reject:**

- The manual annotation needs to be clear.
- The method, experimental setting and evaluation sections require clarification and more details.
- Literature review needs improvement; no literature review for economics data and NER in economics domain is presented. It would be helpful to mention what exists and why it is not appropriate for this paper's analysis.

**Reproducibility:**

4: Could mostly reproduce the results, but there may be some variation because of sample variance or minor variations in their interpretation of the protocol or method.

**Reviewer Confidence:**

4: Quite sure. I tried to check the important points carefully. It's unlikely, though conceivable, that I missed something that should affect my ratings.

**Typos Grammar Style And Presentation Improvements:**

Line 322 remove "label": it is not label, it is the length of the entity itself.

---

> ### Author Rebuttal · Authors · 2023-08-28
>
> We wish to thank the reviewer for his questions and suggestions.
>
> The final paper will include clarifications on the annotation process:
> * The annotation guidelines were produced by experts in the field of economics who decided on the key entities of interest, and experts in NLP and linguistics who designed explicit and unambiguous annotation rules paired with examples (Q1). We will clarify this in the final version of the article.
> * We also computed the inter-annotator agreement and found f1=0.87 and Cohen’s Kappa=0.71 prior to curation (Q3).
> * Annotations underwent a curation phase, where discrepancies between annotators were resolved by a curator who contributed to the guidelines. When the disagreement was due to multiple plausible annotations, a new rule was set and discussed with the team to arbitrate all such cases (Q2).
>
> Additionally, we will add further clarifications following some of the reviewer’s questions:
> * (Q6) the procedure for Table 3  is described in the “NER fine-tuning” portion of the “Models” subsection (2.2, l.222-230), and the splitting strategy (80/20 train-test split and 5-fold cross-validation) is described in the “Sample selection” paragraph of the “Econ-IE dataset” subsection (2.1, l.152-155). We will add cross-references to improve clarity throughout the article.
> * The results in table 3 are F1-scores as mentioned in the second column. The F1-score is computed at the token-level (Q7) as mentioned in section 3.2 (l.261). We will clarify this point in 3.1 in the final version of the draft.
> * We will also clarify what was meant by “isolate entities [X]”, as we meant “compute scores over entities [X]” (Q9). The reviewer asked if having an in-train and an out-of-train entity in the same sentence would be problematic in this case. As our metric is defined at the entity-level and not the sentence-level, the rest of the sentence, including other entities, does not matter.
> * We will clarify that EconBERTa-FS and EconBERTa-FC both outperforms publicly available models (l. 242-243). When comparing EconBERTa-FS to EconBERTa-FC (lines 249-250), we find that the difference is not significant as indicated by the confidence intervals in table 3.
>
> We will also add more information on pre-training:
> * (Q4) Our pre-training data first required extracting raw text from pdfs. We then performed deduplication and noise removal, the quality of which we controlled by checking chunks that were randomly sampled from the clean text. We will detail this process in the final draft.
> * (Q11) The pre-training of the DeBERTa language models took around 60 hours (hardware and hyperparameters in App. A.2 and Table 6), and fine-tuning of the NER models took around 9 minutes (hyperparameters in App. B.4, Table 8).
> * As for the suggestion to use EconBiz papers, it is indeed relevant but greatly overlaps with the data sources we already collected (RePEC, SSRN). We will explore incorporating additional corpora for subsequent releases of the model.
>
> We will also incorporate the reviewer’s suggestion by displaying performance by entity (Q10) and fixing the indexing in figure 3 by starting at an entity length of 1 (Q8).
>
> To conclude, we kindly wish to draw the reviewer’s attention to important aspects of our paper:
> * As we mention in the introduction (l.50-53), to our knowledge there exists no NER dataset in economics, which underlines one of the novelties of our work.
> * Additionally, our main scientific findings (i.e. the results we derived from the series of diagnostic tasks in section 3) are of interest to the community, in addition to the public release of the models and dataset.
> * Finally, we will release the code, model weights, annotated dataset and annotation  guidelines to make all of our results reproducible.

---

### Official Review · Reviewer_GfJT · 2023-08-12

**Soundness:** 4

**Excitement:**

4: Strong: This paper deepens the understanding of some phenomenon or lowers the barriers to an existing research direction.

**Paper Topic And Main Contributions:**

The paper descibes data collection, model training (inlc. pretraining) and deep analysis of a domain specific NER task.
The domain dependend NER task is the detection of  valuable entity types to monitor the impact of policy decisions based on abstracts of economic resarch publications using six enity types.
The modeling not only includes finetunig of three open available Pretrained Language Models, but two pretrained Models by the authors.
These models pretrain a mDeBERTa-V3 model on a corpus of research publications (10.8 B Token).
The authors not only show the performance gain of 3% of the NER-fintuned version of there pre-trained model, but also share insghtful analysis on the NER model peformance reflecting the influence of span length on the model performance.
Additionaly they research the possible influence of lexical and lexicosyntactical model memorization during train time.

**Questions For The Authors:**

Will you share any of your models, code and datasets?
Can you contextualize the analysis from figure 4 with the general class performance ?
I assume that the number of test instances of some combination of entity classes on the one hand and the occurrence of an entity in the training data on the other hand varies a lot.
Is it possible to provide a breakdown of the absolute numbers?
Is it possible to supplement figures 2 (c) and (d) with information on how many unique entities exist in each type (in-train/out-of-train)?
The fact that the distribution of the absolute number of unique entities, at least for entity types like coreference, is very uneven (some concrete entities occur very often, most of them less) is little taken into account. Is it possible to quantify and possibly discuss this fact?
Can the results for table 5 be split by entities in the appendix?
Does the discussed weaker performance of longer entities have a significant impact on the results in Figure 4?
The Figure 6 seems to imply, that smaller models like BERT perform equally well on entities in train set and the performance gain can be explained by better performance on unseen examples. Have you investigated this assumption more closely?

**Reasons To Accept:**

Craft soliede implementation of an approach to tackle a domain specific NER-Task incl. domain specific task independend pretraining, annotation, finetuning, model comparison and research on memorzation tendencies while finetuning foundation models.
The methods for analysing NER models can serve as a blueprint for a better understanding how finetuned foundation models learn to tackle NER tasks in various domains.

**Reasons To Reject:**

No statement about sharing the pretrained model (EconBERTa), or the finetuned versions, or the used code base.
No Interrator agreement reported.
Minor reason: The structure is not matching the typicall structure well: Short related work at the end of the paper, bigger parts of the larged number of references introduced in the introduction.

**Reproducibility:**

4: Could mostly reproduce the results, but there may be some variation because of sample variance or minor variations in their interpretation of the protocol or method.

**Reviewer Confidence:**

4: Quite sure. I tried to check the important points carefully. It's unlikely, though conceivable, that I missed something that should affect my ratings.

---

> ### Author Rebuttal · Authors · 2023-08-28
>
> We wish to thank the reviewer for their appreciation of the paper.
> We plan to make all of our results fully reproducible by releasing the pre-trained EconBERTa models, the fine-tuned versions along with the ECON-IE dataset, as well as all of the code used for the experiments. While we do not have a license to redistribute the raw text of the research publications used for pre-training, we will share a file containing the article names and metadata. All of this material will be openly available on GitHub as indicated in the footnote at the end of page 2.
> Additionally, we computed inter-annotator agreement and found f1=0.87 and Cohen’s Kappa=0.71 prior to curation.
>
> We will also take into account the suggestions to display additional figures. As suggested, we computed the gain averaged across classes in figure 4, and found that it is very similar to all classes except coreference, which also supports our claim that the effect of memorization is stronger for this class.
> Prompted by the reviewer’s suggestions, we also computed two new quantities to account for the memorization effect displayed in figure 4: (i) the proportion of unique in-train entities relative to all unique entities in the test set, and (ii) the mean number of occurrences for each unique entity in the train set only – a refined version of the metric displayed in fig.4c-d normalized by the total number of occurrences for each class. Both metrics more accurately match the observed memorization effect, especially the latter, which also explains the slightly stronger effect measured for population.
>
> We also checked the distribution of length for each entity type to see whether the length of entities (as measured in fig. 3) could have an effect on memorization. While the mean length of coreferences is lower, all entity types have a mean entity length close to 1. The distribution of entity length for each entity type therefore doesn’t seem to account for the memorization effects measured in fig. 4, but it will be an interesting piece of information to display in the appendix.
>
> Finally, we will improve the manuscript structure by moving the “Related Work” section after the introduction.

---

### Meta-Review · Area_Chair_P5QE · 2023-09-22

**Recommendation:** 3

**Metareview:**

Reviewers appreciated the well-written presentation of a new pretrained model and NER dataset for the field of economics. Several reviewers found issues with the level of detail provided about how the dataset was created, as well as the specific protocols for annotation which may impact the reproducibility and soundness of the approach. Annotations were shared with reviewers during the evaluation period, but the pretrained model was not made available. Reviewers also suggested a deeper description of the motivations of targeting the economics domain, domain-specific considerations that express the unique challenges in economics, and additional or revised analysis. Reviewers also suggest a discussion of related work, both for comparing their approach to other domain-specific pretrained LMs, existing, widely-used NER models, and the core tasks in the economics domain. This paper is unlikely to have an impact in NLP communities due to a lack of novel methods or analysis, but may be a valuable contribution for research in economics.

---

### Decision · Program_Chairs · 2023-10-07

**Decision:**

Accept-Findings

**Comment:**

Reviewers appreciated the well-written presentation of a new pretrained model and NER dataset for the field of economics. Several reviewers found issues with the level of detail provided about how the dataset was created, as well as the specific protocols for annotation which may impact the reproducibility and soundness of the approach. Annotations were shared with reviewers during the evaluation period, but the pretrained model was not made available. Reviewers also suggested a deeper description of the motivations of targeting the economics domain, domain-specific considerations that express the unique challenges in economics, and additional or revised analysis. Reviewers also suggest a discussion of related work, both for comparing their approach to other domain-specific pretrained LMs, existing, widely-used NER models, and the core tasks in the economics domain. This paper is unlikely to have an impact in NLP communities due to a lack of novel methods or analysis, but may be a valuable contribution for research in economics.